# Invited Review: Ketosis Diagnosis and Monitoring in High-Producing Dairy Cows

Mariana Alves Caipira Lei [1] and João Simões [2,*]

1  Department of Zootechnics, School of Agricultural and Veterinary Sciences, University of Trás-os-Montes e Alto Douro (UTAD), 5000-801 Vila Real, Portugal; marianalei@utad.pt
2  Department of Veterinary Sciences, School of Agricultural and Veterinary Sciences, University of Trás-os-Montes e Alto Douro (UTAD), 5000-801 Vila Real, Portugal
*  Correspondence: jsimoes@utad.pt; Tel.: +351-259-350-666

**Abstract:** This work reviews the current impact and manifestation of ketosis (hyperketonemia) in dairy cattle, emphasizing the practical use of laboratory methods, field tests, and milk data to monitoring this disease. Ketosis is a major issue in high-producing cows, easily reaching a prevalence of 20% during early postpartum when the negative energy balance is well established. Its economic losses, mainly related to decreasing milk yield, fertility, and treatment costs, have been estimated up to €250 per case of ketosis/year, which can double if associated diseases are considered. A deep relationship between subclinical or clinical ketosis and negative energy balance and related production diseases can be observed mainly in the first two months postpartum. Fourier transform infrared spectrometry methods gradually take place in laboratory routine to evaluates body ketones (e.g., beta-hydroxybutyrate) and probably will accurately substitute cowside blood and milk tests at a farm in avenir. Fat to protein ratio and urea in milk are largely evaluated each month in dairy farms indicating animals at risk of hyperketonemia. At preventive levels, other than periodical evaluation of body condition score and controlling modifiable or identifying non-modifiable risk factors, the ruminatory activity assessment during the peripartum seems to be a valuable tool at farms. We conclude that a technological advance progressively takes place to mitigate the effects of these metabolic diseases, which challenge the high-yielding cows.

**Keywords:** beta-hydroxybutyrate; diagnosis; metabolic diseases; negative energy balance

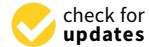



## 1. Introduction

Ketosis is one of the most harmful and damaging metabolic diseases in early lactating dairy cows characterized by high concentrations of circulating ketone bodies which frequently imply productive and reproductive losses, and even death or early culling [1,2]. All these problems represent financial losses that are worsened with the inherent treatment associated with them. In most cases, the treatment of other concomitant diseases, that appear as a cause or an effect of ketosis, further aggravates the financial problem [3].

Knowing that ketosis appears because of an energetic unbalance, it is crucial to invest in prevention, monitoring the most sensible and debile known moments of the productive life of a cow. Even knowing that laboratory tests are more accurate, cowside tests won enlarged utilization over the last few years as they offer real-time results with good accuracy and are simple to perform. Monitorization of milk parameters has been widely studied as a diagnostic method because of its non-invasive sample collection and because these values can be presented on the monthly milk controls. Moreover, they can be used to assess related metabolic diseases such we reported recently [4]. Assessment of body condition and monitoring the thickness of the dorsal fat layer, ruminatory activity, and control of the known risk factors are the most commonly used methods for monitoring ketosis. Understanding that information is the best weapon to success, this review aimed

to highlight the main aspects of the impact and monitoring of hyperketonemia in high-producing dairy cow farms.

Considering a global ketosis prevalence higher than 20% [5], and with estimated costs per case of ketosis that can reach more than €250 [3] in high producing dairy cows, it is imperative to review and update all the available knowledge about ketosis, with particular emphasis on prevention and diagnostic strategies.

## 2. Incidence and Economic Impact of Ketosis

In 2013, Suthar et al. [6] observed that the average prevalence of subclinical ketosis (blood BHB threshold ≥1.2 mmol/L) within ten European countries was 21.8% (ranging between 11.2 and 36.6%) between the 2nd and 15th day of lactation. In 2017, Overton et al. [5] estimated an overall prevalence (total cases per day) of hyperketonemia greater than 20%, corresponding to an overall incidence (new cases on cows at risk per day) of 40%, mainly based on McArt et al. [7] study. In 2018, Brunner et al. [8] showed that the prevalence of subclinical ketosis was around 17% in South Africa, 13.3% in Central and South America, 28.5% in Asia, and 24.85% in Oceania. In fact, hyperketonemia is a major issue on dairy farms which occurs with varying frequency in different countries, as reported in Table 1.

**Table 1.** Ketosis prevalent in several countries of the world.

| Country | Prevalence (%) | | DIM | Method and BHB (mmol/L) Thresholds | Source |
|---|---|---|---|---|---|
| | Subclinical Ketosis | Clinical Ketosis | | | |
| Netherlands | 11.2 | | 5 to 60 | Randox kit—plasma. ≥1.2 | [9] |
| Germany | 20 | 2 | | | |
| Croatia | 14.8 | 1.4 | | | |
| Slovenia | 24 | 2.6 | | | |
| Spain | 22.5 | 2.5 | | Precision Xtra—total blood. Subclinical ketosis ≥ 1.2 | [6] |
| Hungary | 15.6 | 0.4 | 2 to 15 | | |
| Italy | 36.6 | 11.1 | | | |
| Poland | 19.4 | 0.7 | | | |
| Portugal | 29.5 | 6.6 | | | |
| Serbia | 19.5 | 5.7 | | | |
| Turkey | 11.2 | 2.2 | | | |
| Germany | 42 | | | | |
| France | 49 | | | | |
| Netherlands | 48 | | 7 to 21 | Keto-Test—milk. ≥0.1 | [10] |
| Italy | 32 | | | | |
| United Kingdom | 30 | | | | |
| East Canada | 22.6 | | 5 to 35 | FTIR—milk. Suspect: 0.15 to 0.19; Positive: 0.20 | [11] |
| South Africa | 17 | 0 | | | |
| Argentina | 18.8 | 4 | | | |
| Australia | 9.6 | 1.9 | 2 to 21 | Precision Xceed—total blood. ≥1.2 without clinical signs = subclinical ketosis | [8] |
| Brazil | 10.7 | 0 | | | |
| Chile | 14.8 | 2.2 | | | |
| China | 32.9 | 1.2 | | | |

Table 1. *Cont*.

| Country | Prevalence (%) | | DIM | Method and BHB (mmol/L) Thresholds | Source |
|---------|---------------|---------------|-----|------------------------------------|--------|
| | Subclinical Ketosis | Clinical Ketosis | | | |
| Colombia | 8.3 | 0 | | | |
| México | 14.1 | 0.6 | | | |
| New Zealand | 40.1 | 0.1 | 2 to 21 | Precision Xceed—total blood. ≥1.2 without clinical signs = subclinical ketosis | [8] |
| Russia | 14.1 | 0.9 | | | |
| Thailand | 24.1 | 0 | | | |
| Ukraine | 39 | 0 | | | |

BHB—beta-hydroxybutyrate; DIM—days in milk; FTIR—Fourier transform infrared spectrometry.

The proven and known negative financial impact inherent in the appearance and resolution of production pathologies, such as those associated with ketosis, has been presented as one of the main boosters for the investment and implementation of health monitoring programs for the herds [12].

In the United States, the total cost per case of ketosis/year ranges between 129 and US$289 (corresponding to €106.32 and €238.18). According to Liang et al. [13], the value of US$129 comes mainly from the costs inherent to veterinarian interventions and respective treatments in primiparous (68%) and the increased number of days in open in multiparous cows (47%). McArt et al. [1] attributed the US$289 to future reproductive performance losses (34%), death losses (26%), and losses associated with expected milk productivity (26%). In Europe, costs vary between €130 and €257. In his turn, Mostert et al. [2] also considered that 14% of the €130 would be associated with the value lost in discarded milk after the treatment of pathologies associated with ketosis. Raboisson et al. [3] argued that of €257, 80% of the costs were attributable to pathologies associated with ketosis and consequent cull, 11% to losses in milk productivity, and 9% to the increase in open days. Gohary et al. [14] reached C$203 (equivalent to €167.31) for each case of subclinical ketosis, with 13% coming from cull and death, 22% from productive losses, 28% from reproductive losses, and 37% from developing other concomitant pathologies, thus arguing that the largest share comes from direct consequences associated with subclinical ketosis and not from implicit veterinary costs.

Although it is difficult to compare and extrapolate conclusions from studies that used different variables and cost analysis formulas, in most studies analyzed regarding the financial impact of ketosis, it is agreed that the cost for each case of ketosis per year is mainly due to losses of reproductive performance and losses of milk productivity (Table 2).

**Table 2.** Percentage (%) of contribution from different sources of losses to the final cost of a case of ketosis.

| Source | Percentage (%) | | | | Mean (%) |
|--------|----------------|----|----|----|----------|
| Productive losses | 26 | 38 | 11 | 22 | 14.6 |
| Reproductive losses | 34 | 36 | 9 | 28 | 23.6 |
| Losses for associated pathologies | | | 80 | 37 | 23.4 |
| Diagnosis and treatment | 6 | 19 | | | 13.6 |
| Death and cull | 34 | 6 | | 13 | 7.8 |
| Reference | [1] | [2] | [3] | [14] | |

These costs can even double when pathologies commonly associated with ketosis, such as displacement of the abomasum, clinical mastitis, and metritis, laminitis, among others, are present [15]. Moreover, most studies agree that the costs per case of ketosis per year are double or more in multiparous cows than primiparous cows [2,13], although,

McArt et al. [1] reported the opposite. The OR for the development of some pathologies associated with hyperketonemia were gathered in Table 3.

Assigning an average cost of US$289 per case (either primiparous or multiparous), in a prevalence of 21.8% in a herd of 100 cows, the resultant losses would be from 4425 to US$6300 (corresponding to €3646.93 and €5192.24) by a transition period of the herd, depending on the calculation model chosen [16].

**Table 3.** Likelihood (odds ratio) of dairy cows to present different pathologies during postpartum.

| *Pathologies* | OR (CI 95%) | Threshold of Subclinical and/or Clinical Ketosis | *p* Value | Reference |
|---|---|---|---|---|
| Clinical ketosis | 5.4 (3.3–8.8) | >1.4 mmol/L in the blood without clinical signs (A) | <0.0001 | [17] |
| Displaced abomasum | 3.4 (1.9–6.4) | ≥0.1 mmol/L in milk without clinical signs = subclinical ketosis; with clinical signs = clinical ketosis (B) | <0.01 | [10] |
| | 5 (3.5–7.2) | ≥1.2 mmol/L in the blood = subclinical ketosis | <0.001 | [6] |
| | 6.1 (2.3–16.0) | 1.2 to 2.9 mmol/L = subclinical ketosis; ≥3.0 mmol/L = clinical ketosis (blood) (C) | <0.001 | [7] |
| | 3.3 (2.6–4.3) | A | <0.001 | [17] |
| Metritis | 1.5 (1.0–2.0) | B | 0.03 | [10] |
| | 1.5 (1.2–1.8) | ≥1.4 mmol/L in the blood = subclinical ketosis | <0.001 | [6] |
| | 1.8 (1.5–2.0) | A | <0.0001 | [17] |
| Placental retention | 1.5 (1.2–1.9) | A | <0.001 | [17] |
| Mastitis | 1.9 (1.3–2.7) | B | <0.01 | [10] |
| | 1.6 (1.2–2.1) | A | <0.001 | [17] |
| Duplication of SCC | 1.4 (1.3–1.6) | A | <0.001 | [17] |
| Laminitis | 1.7 (1–3.1) | B | 0.05 | [10] |
| | 1.8 (1.3–2.5) | ≥1.2 mmol/L in the blood = subclinical ketosis | <0.001 | [6] |
| | 2.0 (1.6–2.4) | A | <0.001 | [17] |
| Early cull | 1.9 (1.6–2.3) | A | <0.0001 | [17] |
| | 2.6 (1.3–5.2) | 1.2 mmol/L in the blood = subclinical ketosis | 0.008 | [18] |
| | 3.0 (2.2–4.2) | C | <0.001 | [7] |

It is easy to understand how damaging and costly this pathology can be for both the cow and the farmer and how important it is to know how to prevent and early diagnose this disease. That is why the study of the risk factors and the development of new monitoring and diagnostic tests has been the main focus on this subject [19,20].

Even knowing that all cows are, in variable degrees, exposed to negative energy balance, the exact mechanisms that conduct only part of the cows to developing ketosis is not yet fully understood [21], that is why it is still so difficult, in most cases, to perceive with certainty if the concomitant diseases commonly observed in ketotic cows are truly a cause or an effect [18].

### 3. Energy Metabolism in Ruminants

The ruminants' diet contains considerable amounts of structural carbohydrates (cellulose, hemicelluloses, and pectin) and reserve carbohydrates (starch and other water-soluble carbohydrates). Of all these carbohydrates, approximately 90% are degraded in fermentation processes performed by the reticulorumen commensal microbiota; all carbohydrates, except indissoluble fiber—lignin—are a target of the action of ruminal microorganisms.

Only about 10%, corresponding to water-soluble starch or carbohydrates, are digested and absorbed in the small intestine [22].

The degradation of dietary carbohydrates in the rumen occurs in two stages: The first of which involves the digestion of complex carbohydrates into simple sugars, fundamentally through the action of extracellular microbial enzymes; and the second, through the intracellular metabolization of these sugars by microorganisms. This microbial fermentation process results in short-chain volatile fatty acids (VFA)—acetic, propionic, and butyric acid (the most abundant)—as well as in the production of methane and carbon dioxide, which are being eructed [22].

Acetic acid, or acetate, is the main product of the fermentative digestion of carbohydrates in ruminants and is the VFA found in more significant amounts in the peripheral circulation. This can then be used as an energy source, through the Krebs cycle, in various tissues, such as adipose tissue [23]. Still, its main destination is the synthesis of milk fat; several studies prove that dietary supplementation with acetate increases the butter content of milk as it increases lipogenesis in the mammary gland mainly by stimulating the "de novo" synthesis pathways [24–26].

Propionic acid, or propionate, after being produced, passes passively through the ruminal wall (only a small part is converted into lactate upon absorption by the ruminal wall), and from there, the majority is transported to the liver to participate in gluconeogenesis, originating glucose that can be used as an energy source [23]. Propionate represents 90 to 95% of the contribution of VFA to gluconeogenesis [27].

Butyric acid, or butyrate, is converted to BHB, during the absorption process by the rumen and omasum walls, which can then be used as an energy source, through the Krebs cycle [23,28], or as a precursor of milk fat synthesized "de novo" in the mammary gland [29].

To supply the needs of carbohydrates in full, the organism, mainly the liver, dedicates itself to the endogenous production of the 90% of glucose missing, which are not directly supplied by food [30]. Two important processes, gluconeogenesis, and glycogenolysis intervene in the metabolism of carbohydrate synthesis.

Gluconeogenesis has propionate as its main precursor, followed by isobutyrate and valerate [27]; to a lesser extent, this route also uses acetate, lactate, butyrate, non-esterified free fatty acids (NEFA), glycerol (mostly resulting from lipolysis of adipose tissue) [23] and amino acids (especially from Alanine) [31].

As mentioned above, energy can also come from triglycerides (TG) and NEFA in the diet or TG reserves in the body; however, the proportion of lipids in the ruminant diet is generally low. Even so, TG are subjected to the action of bacterial lipases, giving rise to NEFA and glycerol, absorbed by the ruminal wall. On the other hand, long-chain free fatty acids are not directly absorbed by the rumen wall, and therefore, in the small intestine, they are hydrogenated/saturated and hydrolyzed or esterified [22]. These are then transported in chylomicrons to the liver; the TG are hydrolyzed, as described above [23]. In both situations, glycerol enters the glycolysis pathway (in the form of phosphate dihydroxyacetone) for later energy production. The maximum contribution of glycerol to glucose production is observed in the postpartum period, reinforcing its role as an indicator of TG mobilization in adipose tissue. Interestingly, the enzyme responsible for the transformation of glycerol into phosphate-dihydroxyacetone has, until now, only been found in the liver and mammary gland, and therefore, it is expected that only these two tissues metabolize glycerol [27].

Nevertheless, the greatest contribution of TG to energy production comes from the β-oxidation of NEFA (progressive shortening of two carbons in the carbon chain), which culminates in the production of acetyl coenzyme A (Acetyl-CoA) that is sent to the Krebs cycle for obtaining energy [23].

When in more demanding phases, the organism is forced to choose alternative substrates to maintain its essential functions, amino acids can be catabolized to obtain energy. This phenomenon generally occurs in the liver. Again, the final product is acetyl-CoA

which can be sent to the Krebs cycle and converted there into energy. One of the by-products of this process is ammonia, a very toxic compound. Fortunately, much of the ammonia not used by the body to synthesize amino acids is efficiently excreted in the form of urea (in mammals). In ruminants, specifically, part of the urea is recycled via saliva or directly through the rumen wall, depending on the nitrogen levels in the body [23].

## 4. Pathophysiology of Ketosis and Inclusion in the Negative Energy Balance (NEB)

Peripartum is a critical moment for dairy cows. It is marked by profound nutritional, endocrine, metabolic, immune, and reproductive changes [32,33]. Between the 3rd precalving week and the 3rd postcalving week, the transition period [34] occurs, during which the cow experiences a stage of energy deficit [35] that affects health and metabolism, production, and reproduction of the remaining lactation of a cow [36].

It is important to reflect that the last stage of pregnancy can be almost entirely dedicated to the growth of the fetus, so the energy needs to increase [37]. These needs are aggravated by the demand for nutrients for developing the mammary gland that begins to appear at this stage; all these requirements are then aggravated/impaired by the reduction in dry matter intake that occurs at the end of pregnancy [38], the latter mainly due to the compression that the pregnant uterus exerts on the rumen.

Compared to prepartum, at the beginning of lactation, milk production requires an expected energy supply in the diet 30 to 50% higher [16]. These requirements are mirrored by the plasma glucose concentration that decreases after calving, mainly in high producing cows, reflecting the priority energy supply for milk production in the mammary gland; mainly to produce lactose (glucose + galactose), the main osmotic regulator for the mammary secretion of water, thus determining the total volume of produced milk. Effectively, about 85% of the glucose produced at the beginning of lactation is destined for the mammary gland [39].

However, this increased contribution does not occur because, on the contrary, and besides, there is a more pronounced decrease in dry matter intake 24 h before delivery, which only returns to the values prior to this fall 24 h after delivery [40] for the emergence of the sensation of hunger in response to the increased nutritional demand [38]; even so, values equivalent to those of predelivery intake are insufficient to supply these 30 to 50% more energy needed for milk production [16].

This discrepancy between the nutritional demand and the available provisions contributes to the installation of NEB [35].

Given the increased energy requirements for milk production (which peak is usually reached by four weeks) and considering that the maximum intake capacity is only recovered 7 to 8 weeks after delivery, the expected adaptive response (an adaptation of the metabolism of anabolic to catabolic [41]) is generally insufficient to meet these same needs, especially in cows with high productive performance [15].

Being glucose an essential nutrient and inherently associated with the maintenance of the normal vital functions of most tissues and with lactogenesis [15,35], given the insufficient available quantity of it, obtained by gluconeogenesis, in periods of NEB, the body urgently needs to mobilize alternative energy sources—fat reserves in the form of TG; their lipolysis, in adipocytes, results in glycerol and NEFA that are released into circulation and sent to the liver [21].

High concentrations of these circulating fatty acids can impair the insulin signaling pathway, decreasing its sensitivity [42], which in turn exacerbates the mobilization of fat reserves and the entry into circulation of NEFA, thus creating a vicious cycle [43]. As noted by De Koster and Opsomer [44], most studies on the subject confirm that dairy cows go through a phase of insulin resistance between the end of pregnancy and the beginning of lactation; these homeoresis mechanisms are a physiological, adaptive, and transitory phenomenon that aims to prioritize the supply of glucose to the pregnant uterus and the mammary gland over other tissues. Thus, the correct development and survival of the offspring are always the priority.

In the liver, glycerol is used for gluconeogenesis, and NEFA are converted to acetyl-CoA that can have various destinations, such as oxidation to carbon dioxide, oxidation to ketone bodies, hepatic storage in the form of TG, or incorporation into very-low-density lipoproteins, which will be exported as fuel for other tissues. In a demanding phase, such as the beginning of lactation, the number of reserves mobilized is much higher than normal, leading to the production of a large amount of acetyl-CoA. If the Krebs cycle is unable to metabolize the excess acetyl-CoA, this is transformed (oxidized) into ketone bodies [33,45]—acetone, acetoacetate, and BHB [21]—or stored in the liver as TG, which can lead to the fatty liver with negative repercussions on hepatic metabolism [45]. As a result, an increased amount of ketone bodies enters the circulation, which, in a short time, will also occur in milk and urine [38].

## 5. Classification and Forms of Ketosis

### 5.1. Primary and Secondary Ketosis

Ketosis can be classified according to its origin and pathophysiology.

Type I ketosis, also known as primary or described as spontaneous ketosis, corresponds to hyperketonemia in the period between 3 and 6 weeks postpartum, close to the peak of lactation when milk production exceeds the amount of glucose available. In this very demanding phase, glucose precursors coming from the diet (mostly propionate) or muscle protein are insufficient, resulting in a state of chronic hypoglycemia that triggers hypoinsulinemia. In return, lipolysis and ketogenesis pathways are activated [21]. These animals generally do not develop severe hepatic steatosis since, given the high energy requirements, the gluconeogenesis pathway is stimulated to the maximum, and most NEFA are converted into ketone bodies and very little into TG, with no significant accumulation of lipids in the liver [35]. Upon diagnosis, these animals reveal hyperketonemia and hypoglycemia [21].

In contrast, type II, mainly related to secondary ketosis (to other pathologies), occurs at the beginning of lactation, before or at birth, because of the excessive mobilization of adipose tissue [21]. This mobilization results in large amounts of NEFA, which if the gluconeogenesis and ketogenesis pathway are not stimulated to their maximum, are re-esterified in TG. Due to the limited capacity of ruminants to produce low-density lipoproteins in sufficient quantities to export all these TG to other tissues, a large part of them accumulates in the liver, resulting in hepatic steatosis. The high circulation of ketone bodies (although lower than in type I ketosis) [35] cause the body to be resistant to insulin, impairing the use of glucose and starting a vicious cycle. Exaggerated body condition score (BCS) and overfeeding during the dry season are critical risk factors for developing this type of ketosis. At the time of diagnosis, these animals present hyperketonemia accompanied by hyperglycemia [21].

### 5.2. Subclinical and Clinical Manifestation of Hyperketonemia

Hyperketonemia can develop as subclinical ketosis or clinical ketosis [15]. Subclinical ketosis corresponds to hyperketonemia without manifestation of evident clinical signs—the cow maintains the usual appetite without reducing the dry matter intake [21]—and it is verified, according to several authors, at BHB values in the blood (serum) from 1.2 mmol to 1.4 mmol [8], due to the energy demand for dairy production. Clinical ketosis, as the designation suggests, applies to cases of hyperketonemia accompanied by the manifestation of clinical signs and hypoglycemia [46], usually evident at blood BHB concentrations ≥3.0 mmol/L [47].

High concentrations of BHB are not guaranteed to correspond to the depletion of the animal's health status [48]; some animals show clinical signs for slight increases in plasma BHB concentration [49] however, cases without clinical signs associated with BHB concentrations above 3.0 mmol/L are also documented [50].

The high circulating concentrations of TG, NEFA, and ketone bodies in the blood, which characterize ketosis, generate inappetence. BHB, in particular, has the effect of

reducing the signaling of hypothalamic cells responsible for stimulating appetite [51]. Consequently, there is a decrease in food intake and rumen filling/volume followed by anorexia, aggravating the already existing condition [10]. Generally, if they have a choice, the affected animals decrease their intake of concentrate and opt for forages [52].

When associating anorexia with NEB and the excessive mobilization of lipid reserves (which are directly related to the pathophysiology of ketosis), it is expected to observe decreased milk production and loss of BC in affected cows [10].

The feces of the affected animals generally have a drier consistency than those of other animals in the same lactation phase (reminding horse feces), and fur may have a dull, dry, and erect aspect [52].

If the nervous form develops, the cow may lick itself or inanimate objects persistently, exhibit erratic aggressive behavior, present abnormal head posture, and even suffer from blindness. The mechanism that leads to the emergence of nerve ketosis is not yet fully understood [52]. Despite the lack of understanding, Foster [53] suggested that of the three more likely causes for ketosis (hypoglycemia, hyperketonemia, and isopropyl alcohol), increased serum concentrations of isopropyl alcohol are associated with the appearance of nervous signals. This fact was confirmed by another study (Adler et al., 1955 cited by Foster [53], p. 258), where it was demonstrated that the injection of isopropyl alcohol causes clinical signs similar to those of animals with nervous ketosis. Isopropyl alcohol can be produced in the rumen from acetoacetate (and through the rumen wall enter the bloodstream) or in the brain from BHB [53]. Contrary to this, fasting cows which have even lower levels of glucose and similar levels of blood ketones to cows with ketosis, do not show the same signs of nerve ketosis, so hypoglycemia and hyperketonemia do not seem to be the most prevalent factors in triggering nervous ketosis [53].

More often than not, ketotic animals are more apathetic and less active, and some may even manifest ataxia or even the inability to get up. Signs that usually result from the installed hypoglycemia [52].

The fruity odor of ketone bodies in breath and/or milk, mainly caused by ketone [10,52], is another sign which, when present, is not always easy to detect.

According to Dar et al. [54], productive breakdown and selective food intake are the signs most consistently found in cattle with ketosis (Table 4).

**Table 4.** Relative frequency of the different clinical signs that can be manifested by a cow with ketosis (adapted from [54]).

| Clinical Sign | Number of Animals that Showed the Sign | Percentage from the Affected Animals (%) |
|---|---|---|
| Nervous signs | 1 | 4 |
| Reluctance to movement | 1 | 4 |
| Constipation | 4 | 14 |
| Acetone odor on breath or milk | 5 | 18 |
| Dry and fewer feces | 6 | 21 |
| Complete anorexia | 7 | 25 |
| Prostration | 10 | 36 |
| Selective food intake | 21 | 75 |
| Abrupt drop in productivity | 28 | 100 |

Still, all these signs are nonspecific, and some are often not very evident, and therefore, difficult to detect, thereby increasing the risk of obtaining an erroneous diagnosis and the difficulty of correctly distinguishing clinical from subclinical ketosis [15].

Parameters, such as temperature, pulse, and breathing pattern, only deviate from normal if there are other concomitant nosological conditions [52].

## 6. Laboratory Diagnosis and Methods of Monitoring Ketosis

Deviations from metabolic homeostasis are reflected in changes in body fluids, such as blood, urine, milk, and saliva [5]. Of these, the evaluation of some serum metabolites has

been a key point in diagnosing various pathologies, particularly metabolic diseases [50]. Thus, the most common diagnostic method used in diagnosing ketosis has been the analysis of some of these fluids in suspect cows [16].

Concentrations of ketone bodies have been used for diagnosing ketosis in dairy cows for many years [21]. Due to its stability in the blood, BHB has been the ketone body most used in the laboratory diagnosis of ketosis [47], which is considered the "golden standard" method.

The hypothesis of using milk to measure BHB concentration has been increasingly investigated since the analysis and recording of parameters evaluated in milk is already a routine, non-invasive procedure that facilitates monitoring at the herd level. Moreover, unlike blood samples, milk samples reflect the animal's metabolic state for a period of time and not just at the time of harvest [15]. However, the accuracy of the BHB concentration prediction equations in milk has not been sufficiently high to predict the exact BHB concentration. Even so, it has proved useful for monitoring and signaling cows with high concentrations of BHB [55,56].

It is important to emphasize that any instrument and method intended to be used by veterinarians or producers to detect pathologies must be non-invasive, simple to use, and low cost [57].

### 6.1. Laboratory Methods

### 6.1.1. Enzyme Catalysis

This is the traditional test that uses an ultraviolet spectrophotometer or biochemical analyzer to determine blood serum BHB [58]. This method is considered the gold standard, consisting of a colorimetric enzymatic reaction followed by a spectrophotometry analysis [15]. Beyond blood, concentrations of BHB can also be measured in milk using a biochemical analyzer because a statistically significant correlation ($r = 0.705$, $p < 0.01$) was found between BHB in serum blood and milk [59].

### 6.1.2. Fourier Transform Infrared (FTIR) Spectrometry

The use of this diagnostic method began in the late 1990s, and since then, this method has been used by several competent laboratories and organizations around the world to determine the composition of milk (fat, protein, lactose, and urea) monthly in official milk controls, as it allows the rapid processing of many samples, at a low cost per unit, without destructive effects on the milk's composition [48].

Infrared spectroscopy is based on the different rotation of each molecule that determines the energy of the infrared wavelength that is absorbed; the absorbed wavelength is detected by the infrared spectroscope, which then presents the individual absorption spectrum of each molecule, allowing each molecule or substance to be identified by the absorption bands in its spectrum. The combination with the Fourier transformation allowed the mathematical transformation of the interference of the different waves of the entire spectrum of the same substance, at once. With the databases' registration, it is now quick and easy to identify the structure of a substance and compare unknown molecules with those already registered (Rudzik, 1993 cited by Gruber and Mansfeld, 2019 [16], pp. 252–253). Furthermore, technological advances have allowed the electronic storage of spectral information of milk constituents in repeated milk contrasts, which offers the possibility to retrospectively analyze all available information [60].

With this valuable method, concentrations of BHB and acetone can be measured in milk to monitor suspect animals and detect subclinical ketosis sooner/earlier [61].

### 6.1.3. Fluorometry

Larsen and Nielsen [62] recently described this method that measures BHB concentration on milk or blood plasma. The analyses were based on an enzymatic method of BHB oxidation, followed by a second oxidation process of the produced nicotinamide

adenine dinucleotide plus hydrogen (NADH) with resazurin that produces, in proportion, a fluorescent compound, the resorufin.

Larsen and Nielsen [62] verified that the results of this test are not jeopardized by the lack of pretreatment of the plasma or milk samples; this simplifies the process making it able to handle many samples, in-line and in an automated way.

### 6.1.4. Gas-Liquid Chromatography (GLC), Nuclear Magnetic Resonance (NMR) Spectroscopy, and Gas Chromatography-Mass Spectrometry (GC-MS)

Using N-propanol, GLC can be used to determine acetone and BHB in milk or blood serum. On the other hand, NMR and GC-MS can only be used in the blood to determine acetone and BHB values [61].

NMR is a non-destructive, highly accurate, and reproducible technique, but less sensible and less encompassing than MS. The combination of MS with chromatography increases the sensitivity of the technique [63].

All these analyses are subject to delay because they must be performed in a laboratory. As a result, the industry has developed alternative equipment that can be used by producers and that offers results in real-time—the cowside tests [16].

Various types of diagnostic tests are available on the market for use in field conditions. Most of these tests detect acetoacetate or acetone in urine or BHB in milk. The tests are considered semiquantitative since they are based on the color change of the nitroprusside powder (when the sample is added) or of the nitroprusside impregnated in the reactive areas of the dipsticks/strips, which is generally more intense in the presence of higher concentrations of ketone bodies in the sample [64]. Their major components are sodium carbonate, ammonium persulfate, and sodium nitroprusside. Under alkaline conditions, acetoacetate reacts with sodium nitroprusside, resulting in a color change from white to purple that becomes a darker purple color with a higher amount of ketones/degree of ketosis. Apart from the lower sensibilities and specificities compared to the gold standard test, the reliability of these methods is inferior because they rely on a subjective visual interpretation of a color change [61]. However, these tests are usually cheaper, simpler to use, and offer immediate results compared to the previous ones, making them more useful for making/or excluding a clinical diagnosis of ketosis of a suspect sick cow [47].

### 6.2. Cowside Tests

#### 6.2.1. Cowside Urine Tests for Ketosis

Acetoacetate can be evaluated in urine using nitroprusside tablets (Acetest®; Bayer Corp. Diagnostics Division, Elkhart, IN, USA). However, even offering excellent sensitivity, this test usually presents poor specificity (compared to blood tests to assess BHB), making it only useful for evaluating sick cows individually and not for herd monitoring [47].

This same ketone body can be measured with a semiquantitative dipstick (Ketostix®; Bayer Corp., Diagnostics Division, Elkhart, IN, USA) with similar sensitivity and better specificity compared to the previous test [50]. However, these strips are more expensive than other urine ketone tests, making their use on urine impractical [47]. The cost per strip of this test is US$0.20 [65].

According to the manufacturer, urine must be collected from catheterization, spontaneous urination, or urination induced by manual manipulation of the distal urethra [66].

Oetzel [47] believed that some of the obtained false-positive results were due to prolonged contact of urine with the common reagent to these two tests.

Even so, the weakest point of these tests is the difficulty of collecting urine samples comparing to milk samples, making it costly than other alternatives [47]. This point became one of the main limitations of this test because only 50% of the cows could be induced to urinate while sampling [65].

### 6.2.2. Cowside Milk Tests for Ketosis

Assessing acetoacetate: Nitroprusside powder (most common: Utrecht powder; University of Utrecht, Utrecht, The Netherlands and KetoCheck™ powder; Great States, St. Joseph, MO, USA) can be used to assess milk acetoacetate; however, these generally have very poor sensitivity (compared to blood tests to assess BHB) and so, as the previous ones this cannot also be recommended as a reference test for herd monitoring. More important, is due to the poor sensitivity that the value as cowside test for diagnostic decisions for individual cows is very limited.

Even so, they have an advantage towards the urine tests, like an easier collection, ensuring that all the eligible cows for monitorization can be tested [47]. The cost per test ranges from $0.50 to $1.00 [65].

Assessing BHB: This test strip KetoTest™, Ketolac BHB; Sanwa Kagaku Kenkyusho Co., Ltd. (Nagoya, Japan). Detects BHB with satisfactory to good sensibility and specificity, and allows individual diagnostic and herd monitoring, depending on the used cut-point [47]. The cost is $1.70 per test [65].

### 6.2.3. Cowside Blood Tests for Ketosis

Recently, by adapting devices already used in human medicine, the use of portable electronic devices has started. These devices include portable devices and test strips that, in contact with a small amount of blood, develop an electrochemical reaction that determines the concentration of the ketone in the blood [15] with high sensitivity and specificity [67,68]. There are several portable devices for this purpose, notably the Precision Xtra® (from Abbott Diabetes Care, Abingdon, UK), which in addition to analyzing blood samples, also works with milk and urine samples [16]. Another advantage is that this equipment does not require the previous calibration. According to the study by Iwersen et al. [64], in which they evaluated the results presented by this equipment in a test validated with the gold standard method, for a threshold of diagnosis of subclinical ketosis of 1.4 mmol/L of BHB, the measurement with Precision Xtra® in a blood sample offers 100% sensitivity and specificity.

Monitoring using portable BHB meters has, therefore, become one of the methods of choice for detecting ketosis in the field [64,67]. However, although efficient, this is a laborious and expensive method [69]. The cost of the meter is approximately US$30.00, adding approximately US$1.30 per strip [65]. The sensibility and specificity of some of the mentioned tests can be seen in the table below (Table 5).

**Table 5.** Summary table of the characteristics of some ketosis tests (adapted from [16]).

| Test | Measured Substance | Sampled Fluid | Detection Threshold | Sensibility | Specificity |
|---|---|---|---|---|---|
| KetoLac® | BHB | Blood | ≥0.2 mmol/L | 59% | 91% |
| KetoStix® | Acetoacetate | Urine | Cutoff low to moderate | 78 to 49% | 96 to 99% |
| KetoCheck Powder™ | Acetoacetate | Milk | Non defined | 41% | 99% |
| KetoTest™, KetoLac | BHB | Milk | ≥0.1 mmol/L or ≥0.2 mmol/L | 73% or 27% | 96% or 99% |
| Precision Xtra® | BHB | Blood | ≥1.2 mmol/L | 100% | 100% |
| | | Urine | Non defined | 100% | 25% |
| | | Milk | Non defined | 60% | 89% |
| MilkoScan™ FT600 | BHB | Milk | ≥1.2 mmol/L | 80 to 82.4% | 70 to 83.8% |
| | Acetoacetate | Milk | Non defined | 80 to 82.4% | 70 to 83.8% |
| MilkoScan™ FT6000 | BHB | Milk | ≥0.2 mmol/L | 96% | 89% |

According to Oetzel [50], the mammary vein should not be used to collect blood samples, because there the BHB concentration is lower since it is consumed by the udder tissues.

Furthermore, the collection of blood by venipuncture inherently and inevitably, even for a short time, involves the application of containment maneuvers to the animal. Therefore, this technique must be considered as a target of public perception regarding animal management, containment, and welfare [70]. From the above, due to the technical knowledge and skill required to perform the harvests and to the risk of injury to which both animals and humans may be exposed, this procedure should be performed only by veterinarians or other trained personnel, to guarantee fast and minimally stressful harvests [48].

It is also important to reflect that, although blood samples allow for the precise analysis of several parameters [48], a blood sample in itself, only represents the state of the animal at the time of sampling [15]. It is not feasible to carry out frequent and regular harvests for early detection or monitoring of the development of metabolic pathologies, even more, when designed for an entire herd [48].

Given the above, data obtained by non-invasive means, such as milk, will be a better alternative to monitor the metabolism and health of animals. For this, it is a necessary condition that changes verified in the blood are mirrored in the most similar way possible in these alternative means, such as the composition of milk [48].

Knowing that both the health and metabolic state/phase of a dairy cow has an impact on its dairy performance and the quality of the milk, as the metabolic state evolves during lactation, changes in the composition of milk are observed in correspondence [49]. That is why several authors have suggested and agreed that the analysis of milk components is a suitable diagnostic method for detecting ketosis in early lactation [57]. However, analysis of monthly samples limits the detection of pathology or risk assessment to an individual level [69].

### 6.3. Other Available Tests for Ketosis Diagnosis

#### 6.3.1. Fat to Protein Ratio (F:P)

For decades, payment for raw milk in many countries has been based on the concentration of solutes in it, in particular, the fat and protein content. As a result, these parameters generated goals for creating dairy cows that spurred the development of various analytical methods to ascertain this data quickly, inexpensively, and non-destructively of milk. Another advantage of the analysis of milk constituents is the fact that it allows individual daily monitoring of the animals and a reduction in costs through regular harvests [48].

The amount/proportion of fat and protein in milk are variables affected by factors, such as race (genotypes), food management, productive performance, metabolic level, and health status [48]. It is known that, compared to Holstein-Frisian cows, Jersey cows produce less milk, but with a greater amount of fat and protein [71,72]. Similar is expected in the milk of Swiss brown cows [73]. It is also expected that cows which diets are based on hay or rich in oilseeds, or other fat sources, will produce milk with a higher fat content than those with diets based on corn and grain silage in which the fat content in milk decreases. This is because, as has already been verified and demonstrated by several authors, the lipids from the diet are transferred to the lipid fraction of the milk without requiring the energy expenditure associated with its synthesis [74], and ruminants preferentially use acetate in lipogenesis at the expense of glucose.

The expectation is that shortly after delivery, the levels of fat and protein in milk reach the peak/maximum and that thereafter decrease as milk production increases, until the peak of lactation [75]. The reverse evolution will be verified after the maximum production plateau; the amount of fat and protein in milk gradually increases with the decrease in milk production [48].

It can then be assumed that the amount of fat and protein reflects the energy status/condition of a cow. However, during phases of energy deficit, such as the beginning of lactation and the end of lactation, there are increased levels of fat in milk [75] and decreased levels of protein [48].

Having verified that in NEB at the beginning of lactation, the concentration of fat in milk tends to increase, while that of protein tends to decrease, the F:P ratio was, therefore, suggested as a potential indicator of insufficient food energy supply [76].

The increase in the F:P ratio coincides with periods of NEB associated with increased mobilization of lipid reserves; ratios above 1.35 correspond to cows in an energy deficit, according to Gross et al. [75]. The decrease in this is generally associated with lower production of VFA (mainly acetate and butyrate, which are the precursors to the synthesis of fatty acids in the mammary gland) that may incur subclinical ruminal [48].

In 2015, Jenkins et al. [77] studied the thresholds for the F:P ratio and mathematically determined that the optimal cutoff for a diagnosis of subclinical ketosis ([BHB] $\geq$1.2 mmol/L in the blood) would be 1.42 for a sensitivity of 92% and specificity of 65%; however, even if a test with these sensitivity and specificity values was sufficient to monitor a disease, its specificity would result in a high rate of false positives. They then determined that the optimal threshold for greater balance between sensitivity and specificity would be 1.50 for a sensitivity of 75% and specificity of 78%. For F:P ratios > 1.5, Richardt (2004) (quoted by Čejna and Chládek [78], p. 542) observed a 3.5-fold increase in the risk of developing ketosis.

However, King et al. [57] argue that the F:P ratio is not an adequate method for the early detection of hyperketonemia since changes in milk fat only reflect the health status of the cow if it has already suffered or is suffering from the effects of the disease. In his study, despite the strong associations found between the concentration of BHB in the blood and the F:P ratios in milk, no threshold of the F:P ratio showed acceptable sensitivity and specificity to identify hyperketonemia. The authors suggested that the infrequent (ranging from once a week to once a year) and inconsistent calibration of sensors in the automatic milking system that evaluate milk components, both within and between farms, may explain the low accuracy found in the study. On the same line, Kamphuis et al. [79], p. 11 stated that "Calibration is an important prerequisite for good sensor performance, but is often neglected." Furthermore, Fadul-Pacheco et al. [80] observed that milk fat and protein are generally underestimated by the sensors of the automatic milking system, which, as concluded by King et al. [57], may misrepresent F:P ratios.

Another aggravating factor for the imprecision of the F:P ratio as an indicator of hyperketonemia is that the fat and protein content of milk is determined by factors other than the energy balance at the beginning of lactation [57]. The fat content of milk can also reflect, as we saw above, in addition to the level of NEFA [35] and diet, ruminal health (altered, for example, in cases of subclinical acidosis), which can have great effects during this period [81]. Likewise, dietary factors also significantly affect the protein content of milk [82].

Saying this, F:P ratio is a good method for monitoring ketosis at a whole herd level [61].

### 6.3.2. Urea Content in Milk

The interpretation of the urea concentration in milk should always be made in parallel with the concomitant amount of protein in the milk.

Although it would be expected that the increase in milk production would correspond to an increase in the concentration of urea in it since for such production, a greater supply of protein in the diet is indispensable [83]. This generally is not verified because, as previously mentioned, the beginning of lactation is accompanied by a decrease in dry matter intake [84] and the installation of NEB [35].

Therefore, in the first weeks of lactation, low concentrations of protein and urea in milk are observed, indicative of an insufficient supply of energy and food protein, limiting, consequently, the optimum activity of the ruminal flora, which will therefore synthesize fewer amino acids than would be destined for intestinal absorption [48].

On the other hand, high concentrations of urea in milk, coexisting with normal or reduced levels of protein in milk suggest an excessive supply of protein and insufficient energy in the diet, corroborating the need for an adequate and synchronous supply of

energy and nitrogen to the ruminal biota [48]. The deamination of all the protein ingested above the real needs will contribute to the increase of nitrogen production (in the form of ammonia), which is eliminated in the form of urea in urine and milk. This is because nitrogen metabolism requires energy that, at the beginning of lactation, is prioritized for milk production [83].

Given that the concentration of urea in milk shows a very close correlation with its concentration in the blood and given the confirmed association between the protein content in the diet and the concentration of urea in milk, the latter proves to be useful to monitor the supply of protein and energy in the diet and the environmental nitrogen emissions [85].

### 6.3.3. Fatty Acid Profile in Milk

The incorporation of fat in milk is the most expensive process in terms of energy for producing it [48]. Changes in the fat content of milk and even in the proportions of a single fatty acid allow us to assess the metabolic state of dairy cows [86]. From the analysis methodologies available, FTIR technology offers a reliable estimate for the concentrations of saturated fatty acids, monounsaturated, and polyunsaturated fatty acids [48].

The fatty acids (FA) present in milk come from the diet, from the "de novo" synthesis in the mammary gland, from biohydrogenation or bacterial degradation in the rumen, and the mobilization of body fat [86]. The interest in monitoring these compounds arose when producers and consumers realized that they significantly influence nutritional, organoleptic, and other physical-chemical properties, such as oxidative stability during milk processing [48].

The dietary FA (in plants, most correspond to long-chain unsaturated fatty acids) are incorporated into milk soon after its absorption. From the process lipolysis in adipose tissue, mainly during NEB, arrive at the mammary gland mainly long-chain fatty acids (saturated and unsaturated) and from lipogenesis of non-lipid precursors, such as carbohydrates, (lipogenesis "de novo") in the mammary gland result short and medium-chain fatty acids [48]. However, Palmquist et al. [87] found that long-chain FA from plasma is incorporated into milk fat and inhibited the "de novo" synthesis of short and medium-chain FA in the mammary gland.

From the above, it is correct to verify that the proportion of short and medium-chain fatty acids is lower in early lactation, increasing later [86], while the proportion of monounsaturated FA, especially oleic acid, decreases with the restoration of the energy balance after delivery [88]. Gross et al. [88] observed similar changes in the composition of FA in milk that occurred in just a few days after the reduction of nutrient input, due to food restriction; the fluctuation of the saturated and monounsaturated FA classes reflected the energy deficit.

Van Haelst et al. [89] evaluated whether the concentrations of a specific FA (oleic acid) in milk fat would be suitable for the early detection of subclinical ketosis since lipolysis of adipose tissue precedes developing ketosis. High concentrations of this FA proved to be adequate for predicting the occurrence of subclinical ketosis, mainly because oleic acid was high in milk fat before the detection of ketosis. However, we cannot disregard the intervention of the diet in the FA composition of milk. Supplementation of fat, such as oleic acid, attenuates the energy deficit by increasing the concentration of milk fat and the proportion of FA. Thus, changes in the composition of FA in milk should be interpreted with caution, always taking diet into account [48].

Recent studies have shown that parameters measured in milk other than BHB, such as the predicted NEFA concentration in the blood and the FA concentration synthesized "de novo", when analyzed at any time during the first 18 days of lactation are associated with an increased risk of developing pathologies or removing the animal from the herd in the first 30 days of lactation [69].

In ruminants, BHB is used as a precursor to the "de novo" synthesis of fatty acids in the mammary gland. In their study, Zhang et al. [29] have demonstrated that high BHB values (1.2 and 2.4 mmol) induce lipogenesis in mammary epithelial cells in vitro, insofar

as they induce the activation of lipogenic transcription factors and increase the expression of lipid synthesis enzymes.

### 6.3.4. Blood NEFA

While the concentration of BHB increases when the liver can no longer compact with the excessive supply of NEFA, which is why they are converted into ketone bodies, the concentration of NEFA reflects the energy need of the animal that is helping by mobilizing lipid reserves [90]. It is important to know that BHB and NEFA are not elevated in the same animal at the same time [65].

That is why, several authors verified that cows with blood NEFA concentrations above the thresholds, antepartum and/or in early postpartum, were at higher risk of ketosis [45,91,92].

Therefore, the concentration of NEFA is a good indicator of the level of mobilization of body lipid reserves and is, therefore, more useful for monitoring in the predelivery period, while the BHB concentration, as an indicator of the extent/level of ketogenesis, becomes most useful in the postpartum period [93].

Even NEFA was found to be a better predictor of future negative outcomes, and its values can be more stable than BHB throughout the day; NEFA values are more susceptible to misreading, due to stress at sampling and inadequate sample manipulation. Furthermore, processing blood or serum samples for NEFA testing is limited by requiring a diagnostic laboratory and the cost per sample is approximately $11.00

In opposition, BHB is easier (there are cowside tests available) and cheaper to measure accurately. That is why BHB should be the first choice for monitoring, and when there is evidence that transition cows of a herd are experiencing any problems without significant elevation in BHB, NEFA should be evaluated [65].

### 6.3.5. Analysis of Exhaled Breath

In 1996, Dobbelaar et al. [94] observed that ketosis could be diagnosed by analyses of the acetone concentrations in the exhaled breath. This concentration was correlated with concentrations of serum BHB ($r = 0.81$) and milk acetoacetate + acetone ($r = 0.70$).

Following studies in cattle and small ruminants, though not focused on the study of ketosis, confirmed the possibility to measure ketones' concentration on a sample of exhaled breath [95]. Inclusively, Küntzel et al. [95] state that they "successfully introduced a technical setup to collect and analyze exhaled breath from adult cattle" [95], p. 450. Combining two different analytical techniques, they were able to assess breath composition and identify the present volatile organic compounds. They also believe that the sampling setup would probably be appropriated for field trials.

### 6.3.6. Metabolomics

It is a field of science in development that detects and quantifies metabolites and small molecules like ketones, fatty acids, sugars, amino acids, nucleotides, organic acids, etc. [63]. The elaboration of metabolites' profiles in biofluids, cells or tissues, allows the discovery of new pathologies biomarkers [96].

Metabolomic analyses use one or more analytical techniques, usually NMR spectroscopy and mass spectrometry (MS), two non-destructive, highly accurate, and reproducible techniques. In their review article, Tran et al. [63] found some studies regarding bovine ketosis. These studies found "differential metabolites among groups, thereby providing information on pathogenesis, early diagnosis, and prevention" [63], p. 7 of ketosis in dairy cows and some "Results suggests disrupted metabolic pathways in ketosis (fatty acid and AA metabolism, glycolysis, gluconeogenesis, and pentose phosphate pathway)." [63], p. 7. Even having some limitations, this new scientific field has an enormous potential that has been being underutilized in veterinary medicine.

## 7. Control and Prevention

In addition to the methods of monitoring ketosis described in the previous point, it is important to identify some procedures and a set of modifiable and non-modifiable risk factors that have an impact on the control and prevention of ketosis.

### 7.1. Ruminatory Activity

Mann et al. [97] clarified that, compared to those that receive diets that provide energy well above maintenance needs, cows fed in the dry period with diets with restricted/controlled energy, have a lower risk of being affected by ketosis in the postpartum period, without prejudice to milk production. These animals manifested a less deep NEB and episodes of ketosis to a lesser extent and number.

The study by Kaufman et al. [98], suggested that the monitoring of ruminatory activity during the peripartum, for example, with the use of rumination collars, may contribute to the individual and timely identification of pathologies of the initial stage of lactation. In this study, they found that multiparous cows that had reduced rumination time (by about $25 \pm 12.8$ min/day less than healthy cows between the 2nd week before delivery and the 4th week after giving birth) in the week immediately before giving birth were more likely to be affected by ketosis. Moreover, those who manifested this reduction (approximately $44 \pm 15.6$ min/day less than healthy cows between the 2nd week before delivery and the 4th week after delivery) in the week immediately after delivery would be more probably affected, not only by ketosis, but also by another pathology of early lactation. In the same year, Schirmann et al. [99] testified that cows with postpartum ketosis (BHB $\geq 1.2$ mmol/L) spent 14% less time ruminating during precalving (considered in the study, 10 days before calving) than healthy cows.

### 7.2. Assessment of Body Condition and Monitoring the Thickness of the Dorsal Fat Layer

Monitoring the BCS of cows in the precalving period had proved to be a useful tool in managing the health of the herd [100,101], because Busato et al. [102] found that high scores in the assessment of BCS (>3.25) before parturition were associated with a high loss of body mass in the first postpartum weeks since these animals experienced higher rates of fat mobilization; this phenomenon has been proven by the increased concentrations of circulating NEFA and BHB. Gillund et al. [100] and Roche et al. [101] verified the same phenomenon. Still, Busato et al. [102] found that the metabolic state is optimal in animals of BCS = 3.25 in case they do not lose much body condition in the postpartum period ($\Delta$BCS between prepartum and eight weeks postpartum $\leq 0.75$).

For this monitoring, producers can use the Edmonson BCS classification table (scale 1—emaciated to 5—severely over-conditioned, with increases of 0.25 point) (Edmonson et al. [103]) based on which Gillund et al. [100] recommended a BCS score of <3.5 at delivery to prevent massive fat mobilization. If each animal is evaluated and this evaluation recorded at least once, the producer can, from then on, regularly check changes in the BCS [16].

On the other hand, the method of measuring the thickness of the subcutaneous fat layer on the back using ultrasound necessarily involves the intervention of a veterinarian, which generally makes it more laborious; however, the results obtained are more objective and precise, and therefore, its implementation is recommended to collect measurements/results comparable to those obtained by assessing body condition within the same herd [16]. The method involves measuring the thickness of the subcutaneous fat layer on the back, accumulated between the skin and the deep trunk fascia, which presents itself as a white line hyperechogenic to ultrasounds, after discounting 5–6 mm for the dermis. The sacrum region is the place of choice for this measurement because it is where the largest reserve of adipose tissue on the back is gathered; for said measurement, a horizontal line should be imagined between the ischial tuberosity and the coxal tuberosity and join the fourth to the fifth caudal part of that line, vertically, to the junction of the sacrum with the first caudal vertebra. Due to the high correlation between the amount of dorsal fat and the body fat

content, we can assess the body condition of each animal from the first (Staufenbiel, 1992 cited by Schröder and Staufenbiel [104] (pp. 5–6); [16]).

Although these two methods seem practical and simple, the tendency to increase the size of herds discourages producers from regularly monitoring these variables, given the time and workforce required [16].

### 7.3. Risk Factors of Ketosis

To prevent the emergence of ketosis, the study of risk factors and the most appropriate nutritional management has been the subject of discussion in various parts of the globe [19,20]. The most prevalent and consistently referred risk factors are increased parity (number of lactations), high concentrations of NEFA in the predelivery period, and elevated body condition [5]. Vanholder et al. [20] also add the birthing season, duration of the dry period, duration of the previous lactation, and liters of colostrum produced as risk factors for developing hyperketonemia.

### 7.3.1. Breed

The prevalence of ketosis is generally higher in Jersey cows (19%) (ranging from 11.4% to 25% in herds of the breed) than in Holstein cows (14%), where the prevalence in herds can range from 0 to 28% [105]. Biswal et al. also verified that the incidence of ketosis in crossbred Jersey cows was not only higher than in crossbred Holstein Friesian, but also higher than the incidence of this disorder in local indigenous breeds (Odisha state, India) [106].

In a study with a more diverse sample, Erb and Martin determined that the OR for the occurrence of ketosis on Holstein, Ayrshire, and Guernsey cows was 1.57, 1.68, and 0.61, respectively [107].

In 1985, Andersson and Emanuelson verified that Swedish Red and White cows had higher milk acetone values than Swedish Friesians cows, agreeing with the higher prevalence and incidence of ketosis in the first compared to the last breed, registered by several previous authors [108], Bendixen et al. came to the same conclusion later, in 1987 [109].

### 7.3.2. Days in Milk

Since hyperketonemia is the main repercussion of postpartum NEB, most studies are expected to report a higher prevalence of ketosis in the first two weeks of lactation, a period after which there is a significant decline in prevalence [9,11].

On the same line, McArt et al. [7] reported having detected the peak incidence (22.3% of cows with a positive first test) and prevalence (28.9% of cows with a positive test) of ketosis observed on the 5th day of lactation, and van der Drift et al. [9] found a reduction in the prevalence of ketosis from 16.8 to 7% between the 1st and 2nd month of lactation.

### 7.3.3. Parity (Number of Lactations)

A considerable number of authors have suggested that there is a direct relationship between increased parity and the occurrence of ketosis [105,106]. However, Chandler et al. [105] have verified the reverse in Jersey cows, that the prevalence is higher in primiparous. Santschi et al., 2016 [11] found increasing cumulative prevalence, from 18.8% to 27.6% from the first to the third lactation or higher.

Several authors, such as Berge and Vertenten [10] and Kaufman et al. [98], argue that the risk of developing ketosis or being rejected because of the diagnosis is higher in multiparous cows, due to higher productivity and possible problems faced in the previous lactation and drought period.

However, even though Rathbun et al. [110] have detected an increase from 8.6% to 22.2% of the cumulative incidence of ketosis from the first to the third lactation or higher, they argue that the emergence of ketosis is due to the energy demands of the present lactation and not to the lactation productivity past or genetic potential for milk production.

Thus, according to Miettinen et al. (1992) (cited by Miettinen and Setälä [111], p. 6), the most appropriate will be to conclude that ketosis is more likely to occur in high productive performance cows, due to the high demand for energy and nutrients for effect.

### 7.3.4. Birth Season

Spring is generally indicated by most authors as the most problematic season for the occurrence of ketosis [15].

Nevertheless, the evidence is not sufficient to state outright that the prevalence is higher in the spring [11], with other authors even advocating a higher prevalence in the winter, others in the fall [11,20], and others in the summer [9].

Vanholder et al. [20] believe that the lower quality of the silage used in the first half of the year may explain the phenomenon.

### 7.3.5. Effective Size

Given that, in larger herds, strategies, such as creating lots/groups according to milk productivity, are generally implemented and allow for a better nutritional need supply management for each group. The prevalence of ketosis in these is generally lower than in herds of smaller dimensions [10].

### 7.3.6. Housing

In 2014, Berge and Vertenten [10] found that the prevalence of ketosis was lower in animals housed in cubicles, cubicles, and free parking spaces or cubicles with dividing bars than in systems with free parking spaces with straw floors, and even lower than in the free-grazing system.

### 7.3.7. Nutrition

The nutritional management of pregnant heifers, dry cows, and postpartum cows at the beginning of lactation is decisive for the prevalence of ketosis in these animals [11].

The differences in the prevalence of ketosis that occurs, due to different dietary habits are mainly because of the difficulty of assessing the nutritional intake and health status of animals under grazing or accommodated in a straw yard [15].

For these reasons, the prevalence of ketosis is generally lower in herds fed with forage and concentrate separately or fed only with concentrate mix than in herds partially fed with concentrate mix, that is, fed with this concentrate between periods [10].

## 8. Conclusions

Measurement of blood BHB values in serum or plasma is still recognized by several studies as the gold standard test for diagnosing ketosis. However, with the development of science and new devices, it is important to prioritize developing highly sensitive and specific tests that require minimal manipulation and stress of the animals. Thus, we would have an easier process with reduced risk for all the intervenients. Even more, existing already good devices for individual monitoring, it is important to invest in herd monitoring tests, that can avoid the costs and human recourses that would be necessary to monitor all suspect animals with individual tests. That is impracticable, frustrating, and leads producers to give up on monitoring. That is why milk analyses seem so promising: They are easy to collect with no additional stress, they are in line with the farms' milking routines, can be performed in all lactating animals, and previous values can be saved, allowing the creation of an evolutive profile of the variables, that can help farmers to early predict ketosis.

However, it must not be forgotten that despite various methods and indicators enabling the detection of ketosis, precise feeding of cows is still important, especially in the transition period and in the first stage of lactation, affecting the cows' body condition before delivery.

These are still the main factors influencing the metabolic status and level of blood biochemical indices and the composition of cow's milk.

It is then important to deepen the already available or to develop/discover new preventive strategies. For that to happen, it is needed to keep studying the pathophysiology and risk factors of ketosis.

Only with the prevention, monitoring, and early diagnosis will we fight this pathology, which treatment and consequences can be so costly to the farmers.

**Author Contributions:** Conceptualization, M.A.C.L. and J.S.; data curation, M.A.C.L. and J.S.; writing—review and editing, M.A.C.L. and J.S.; visualization, M.A.C.L. and J.S.; supervision, J.S. All authors have read and agreed to the published version of the manuscript.

**Funding:** This research received no external funding.

**Institutional Review Board Statement:** Not applicable.

**Informed Consent Statement:** Not applicable.

**Conflicts of Interest:** The authors declare no conflict of interest.

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
