# Peer review of "Invited Review: Ketosis Diagnosis and Monitoring in High-Producing Dairy Cows"

_2624-862X, doi:10.3390/dairy2020025_

Round 1

Reviewer 1 Report

This review paper has a great scientific and practice merit  and has written in high style and in good ways describe a coplex problems of ketosis in dairy herds which can produce a large economic loses as consequence  healing cost, high culling rate, low fertility, low immunity  and frequent appearance of banal infections, etc).

This review involve a important area in prevalence, diagnosis. control and prevention of ketosis (Incidence and economic impact of ketosis, energy metabolism in ruminants, pathophysiology of ketosis and inclusion in the negative energy balance (NEB), classifications and forms of ketosis,  laboratory diagnosis and methods of monitoring ketosis, control end prevention).

Especially is important description of cheap diagnostic methods in row milk for monitoring ketosis in the dairy herds as fat : proteim ratio, urea content, fatty acid profile in milk and etc.

And finally, this paper is very good contribution in complex problems with ketosis in dairy cows and very usefull for scientis, veterinian and farmers.

Suggestion to authors: to check values (Cut point) for BHB in blood and milk for subclinical end clinical ketosis in Tables 1, 3 and 5.

Cut point for BHB for subclinical ketosis is 1,2 or 1.4 mmol/l, and > 3.00 mmol/l for clinical ketosis and about 10x lower for BHB  in milk.

Author Response

Thanks for your considerations.

Suggestion to authors: to check values (Cut point) for BHB in blood and milk for subclinical end clinical ketosis in Tables 1, 3 and 5.

Answer: All values of Cut points were checked, and the changes highlighted in the tables.

Reviewer 2 Report

Ketosis is still a significant problem for most herds of high-yielding cows. The review presented for evaluation covers issues related to the diagnosis and methods of detecting cow ketosis.

In the study, based on 107 items of literature, the authors discussed the incidence and economic effects of ketosis, energy metabolism, a significant role of the negative energy balance in the first period of lactation, forms of cow ketosis, as well as the diagnostic methods used to detect the state of ketosis in cows (laboratory tests, cowside tests), are presented.

All indicated methods (laboratory and field) for determining the level of ketosis rates in cows, depending on the technical feasibility, potential costs and accuracy, can be used for early detection and effective prevention measures.

Some coments:

One cannot fully agree with the statement that “However, this approach is only useful to diagnose a minority of subclinical ketosis since most animals will not show clinical signs and therefore will not be tested”. After all, early detection of subclinical ketosis is important from the point of view of reducing and preventing its occurrence. This section indicates the usefulness of the BHB level or acetone in milk for the detection of ketosis during lactation.

In table 1: Serva, Hungry, Polonia – rather: Serbia, Hungary, Poland;

Line 573: “…butter content of milk…”, rather “fat content”;

In subsection 7.3 you can omit the next short subsections and combine them into a whole, especially that in the first paragraph of this section have already been given some of the factors favoring the occurrence of ketosis.

Line 730-732: When discussing the influence of the breed of cows, only two dairy breeds were mentioned, are there no results of studies in the herds of other breeds in the available literature?

Line 786: the term golden standard for BHB levels can be found in many studies;

In the conclusions: however, it should be emphasized that despite various methods and indicators enabling the detection of ketosis, precise feeding of cows is still important, especially in the transition period and in the first stage of lactation, and consequently affect the level of cows' condition before delivery. These are still the main factors influencing the metabolic status and level of blood biochemical indices and the composition of cow's milk. Maybe it is worth including this statement?

Author Response

Thanks for your considerations.

One cannot fully agree with the statement that “However, this approach is only useful to diagnose a minority of subclinical ketosis since most animals will not show clinical signs and therefore will not be tested”. After all, early detection of subclinical ketosis is important from the point of view of reducing and preventing its occurrence. This section indicates the usefulness of the BHB level or acetone in milk for the detection of ketosis during lactation.

Answer: The word “minority” was misapplied. This sentence was removed.”

In table 1: Serva, Hungry, Polonia – rather: Serbia, Hungary, Poland;

Answer: Corrected.

Line 573: “…butter content of milk…”, rather “fat content”;

Answer: Corrected.

In subsection 7.3 you can omit the next short subsections and combine them into a whole, especially that in the first paragraph of this section have already been given some of the factors favoring the occurrence of ketosis.

Done.

Line 730-732: When discussing the influence of the breed of cows, only two dairy breeds were mentioned, are there no results of studies in the herds of other breeds in the available literature?

Answer: Of course, but ketosis is a “production disease”, more relevant for high-yielding cows. We reported two other dairy breeds (Swedish Red and White cows), L740-743.

Line 786: the term golden standard for BHB levels can be found in many studies;

Answer: L324-325: we added “…which is considered the “golden standard” method.”

In the conclusions: however, it should be emphasized that despite various methods and indicators enabling the detection of ketosis, precise feeding of cows is still important, especially in the transition period and in the first stage of lactation, and consequently affect the level of cows' condition before delivery. These are still the main factors influencing the metabolic status and level of blood biochemical indices and the composition of cow's milk. Maybe it is worth including this statement?

Answer: L812-816: “Yet, it must not be forgotten that despite various methods and indicators enabling the detection of ketosis, precise feeding of cows is still important, especially in the transition period and in the first stage of lactation, affecting the cows' body condition before delivery.

These are still the main factors influencing the metabolic status and level of blood biochemical indices and the composition of cow's milk.”

Reviewer 3 Report

Ketosis is a serious problem in dairy farming around the world, which is why expanding knowledge about it is so important. The presented review wonderfully collects the current knowledge regarding the diagnosis of this pathology. The work is written in a clear, logical way and it is very pleasant to read. It will be an interesting article presenting the latest knowledge on ketosis and its diagnosis.

Author Response

Thanks for your considerations.